# Association of Skeletal Muscle and Adipose Tissue Distribution with Histologic Severity of Non-Alcoholic Fatty Liver

**DOI:** 10.3390/diagnostics11061061

**Published:** 2021-06-09

**Authors:** Min-Kyu Kang, Jung-Hun Baek, Young-Oh Kweon, Won-Young Tak, Se-Young Jang, Yu-Rim Lee, Keun Hur, Gyeonghwa Kim, Hye-Won Lee, Man-Hoon Han, Joon-Hyuk Choi, Soo-Young Park, Jung-Gil Park

**Affiliations:** 1Department of Internal Medicine, College of Medicine, Yeungnam University, Daegu 42415, Korea; kmggood111@naver.com (M.-K.K.); tubaekj@gmail.com (J.-H.B.); 2Department of Internal Medicine, School of Medicine, Kyungpook National University, Kyungpook National University Hospital, Daegu 41944, Korea; yokweon@knu.ac.kr (Y.-O.K.); wytak@knu.ac.kr (W.-Y.T.); magnolia1103@naver.com (S.-Y.J.); deblue00@naver.com (Y.-R.L.); 3Department of Biochemistry and Cell Biology, School of Medicine, Kyungpook National University, Daegu 41944, Korea; KeunHur@knu.ac.kr (K.H.); aoet111@gmail.com (G.K.); 4Department of Pathology, School of Medicine, Keimyung University Dongsan Hospital, Daegu 42601, Korea; hwlee@dsmc.or.kr; 5Department of Pathology, School of Medicine, Kyungpook National University, Kyungpook National University Hospital, Daegu 41944, Korea; one-many@hanmail.net; 6Department of Pathology, College of Medicine, Yeungnam University, Daegu 42415, Korea; joonhyukchoi@ynu.ac.kr

**Keywords:** non-alcoholic fatty liver disease, non-alcoholic steatohepatitis, sarcopenia, adipose tissue, skeletal muscle, body composition

## Abstract

Adipose tissue and skeletal muscle is associated with non-alcoholic fatty liver disease (NAFLD). This study evaluates the association between body composition and histologic severity in patients with NAFLD. Using the cross-sectional CT images at the level of L3 vertebra and the histologic findings of 178 patients with biopsy-proven NAFLD, we analyzed the correlation of the histologic findings to the skeletal muscle index (SMI), subcutaneous adipose tissue index (SATI), and visceral adipose tissue index (VATI), which is defined as the body composition area (cm^2^) by height squared (m^2^). The clinical and laboratory features with body composition were analyzed to determine the risk factors for advanced fibrosis. The VATI significantly increased in severe non-alcoholic steatohepatitis (NASH) or advanced fibrosis. In addition, the VATI was correlated with the NAFLD activity score (NAS) and the fibrosis stage. In multivariate analyses, age (odds ratio (OR), 1.09; 95% confidence interval (CI), 1.02–1.19; *p* = 0.025), severe NASH (OR, 8.66; 95% CI, 2.13–46.40; *p* = 0.005), and visceral adiposity (OR, 6.77; 95% CI, 1.81–29.90; *p* = 0.007) were independently associated with advanced fibrosis in patients with NAFLD. Visceral adiposity is correlated with the histologic severity of NAFLD, which is independently associated with advanced fibrosis.

## 1. Introduction

The incidence of non-alcoholic fatty liver disease (NAFLD), one of the most important chronic liver diseases, has been increasing in obese people over the past three decades [1]. In Western countries, approximately one-fourth of the population is affected by NAFLD, which has led to an increase in healthcare costs [2]. The spectrum of NAFLD ranges from simple steatosis to non-alcoholic steatohepatitis (NASH), which leads to cirrhosis and hepatocellular carcinoma [3]. Several epidemiological studies have reported associations between NAFLD and metabolic syndrome (MetS), type 2 diabetes, obesity, and cardiovascular diseases [4,5]. Recently, as a variable clinical manifestation, fat accumulation in the visceral organs harboring genetic polymorphism was considered as a subtype of NAFLD without obesity [6]. Regardless of obesity, visceral adipose tissue (VAT) is associated with MetS and T2D [7]. Specifically, VAT is associated with the degree of hepatic fat infiltration, even in non-obese adults [8]. A positive energy balance induces fat accumulation in the subcutaneous adipose tissue (SAT), which has a relatively lesser effect on insulin resistance (IR) at the initial stage of NAFLD [9]. However, when adipose tissue dysfunction with intolerance of energy excess develops, fat is accumulated in the visceral organs, including the liver, heart, skeletal muscle, and VAT [10]. Considering that blood flows from VAT and drains into the liver through the portal vein, high concentrations of portal free fatty acids and cytokines secreted from VAT adipocytes may contribute to NAFLD and IR [11]. Additionally, crosstalk between the liver, adipose tissue, and pro-inflammatory molecules, such as interleukin-6 (IL-6) and tumor necrosis factor-α, released from activated macrophages and adipokines plays a pivotal role in NAFLD progression [12].

The skeletal muscle is the primary site of glucose disposal, which is stimulated by exercise and crosstalk between insulin and the pancreas [13]. With the development of sarcopenia, defined as the age-dependent loss of skeletal muscle, the risk of MetS, NAFLD, and cardiovascular diseases increases [14,15,16]. Specifically, sarcopenia is an independent risk factor for the progression of NAFLD, including NASH and advanced fibrosis, regardless of obesity and IR [17]. Therefore, the development of visceral obesity and sarcopenia could be independent pathogenic mechanisms of NAFLD. Although several studies have reported on the association of these two risk factors with NAFLD, most of these studies evaluated this association as an indirect measurement of body composition or using noninvasive methods for NAFLD diagnosis, leading to potential confounders.

This cross-sectional study investigated the association between the histologic features, skeletal muscle mass, and adipose tissue distribution using computed tomography (CT) scans in patients with NAFLD.

## 2. Methods

### 2.1. Patients

Data of patients who underwent an ultrasound-guided percutaneous liver biopsy for suspected NAFLD were collected consecutively at two tertiary hospitals retrospectively between January 2013 and December 2019. The exclusion criteria were as follows: chronic viral hepatitis, significant alcohol consumption (male > 140 g/week; female > 70 g/week), evidence of drug-induced liver injury, secondary fatty liver caused by drugs or hereditary diseases, and the unavailability of a CT scan within 1 month based on medical records. This study was reviewed and approved by the Institutional Review Board of Yeungnam University Hospital (IRB No. 2020-03-009) and the requirement of obtaining written informed consent from the patients was waived due to the retrospective nature. This study was conducted in accordance with the principles of the Declaration of Helsinki, and it was not possible to involve patients or the public in the design, conduct, reporting, or dissemination plans.

### 2.2. Histopathological Evaluation

A single experienced pathologist at each hospital independently reviewed all of the specimens at each hospital. NAFLD was defined as excessive hepatic fat infiltration of greater than 5%, determined by the histology, without evidence of other liver diseases [18]. NASH was defined as a combination of the presence of steatosis, lobular inflammation, and ballooning degeneration according to the current guidelines [18,19]. The NAFLD activity score (NAS) was evaluated according to the system devised by the Pathology Committee of the NASH Clinical Research Network [20]. Steatosis, lobular inflammation, and ballooning degeneration were scored on 1–3, 0–3, and 0–2 scales, respectively. A NAS > 4 was defined as severe NAFLD. The fibrosis stage was estimated using a 5-point scale according to the Kleiner scoring system as F0–F4, and advanced fibrosis was defined as stages F3 and F4 [20].

### 2.3. Assessment of Skeletal Muscle Mass and Adipose Tissue Distribution

The area of skeletal muscle mass and adipose tissue distribution were evaluated based on the cross-sectional area at the level of the third lumbar (L3) vertebra, which shows a high correlation with whole-body skeletal muscle mass and adipose tissue distribution, on abdominal CT images using a Picture Archiving and Communications System (Centricity, GE Healthcare) [21,22]. Abdominal CT images for body composition were analyzed using MATLAB version R2014a (MathWorks Inc., Natick, MA, USA). The cross-sectional areas of each body composition at L3 were calculated using an open-source software program (BMI_CT, https://sourceforge.net/projects/muscle-fat-area-measurement/) (accessed on 27 November 2017) with standard Hounsfield unit thresholds of −29 to 150 for the skeletal muscle and −190 to −30 for the visceral fat [23]. The area of the skeletal muscle mass was defined as a combination of the areas of the psoas, paraspinal, transversus abdominis, rectus abdominis, and external and internal oblique muscles. The area of the SAT was estimated as the boundaries of the skeletal muscle and line of the abdominal skin (Appendix A). The three-body composition indexes (cm^2^/m^2^), including the skeletal muscle index (SMI), SAT index (SATI), and VAT index (VATI), were defined as the body composition area (cm^2^) by height squared (m^2^). Sarcopenia was defined as an SMI < 50 cm^2^/m^2^ in men and an SMI < 39 cm^2^/m^2^ in women [24].

### 2.4. Assessment of Clinical and Laboratory Variables

Anthropometric measurements, including height, weight, and seated blood pressure (BP), were measured and recorded by trained hospital staff. Obesity was defined by a body mass index (BMI) ≥ 25 kg/m^2^, according to the Asia Pacific region criteria [25]. Diabetes mellitus was defined as a fasting plasma glucose (FPG) level ≥ 126 mg/dL and antidiabetic medication use [26]. Hypertension was defined as (i) a systolic BP level ≥ 140 mmHg, (ii) a diastolic BP level ≥ 90 mmHg, or (iii) antihypertensive medication use. The following routine laboratory variables were assessed within 1 week of liver biopsy: FPG, serum aspartate aminotransferase (AST), alanine aminotransferase (ALT), γ-glutamyltransferase, blood urea nitrogen, creatinine, high-sensitivity C-reactive protein, and lipid profile.

### 2.5. Statistical Analyses

Quantitative data are expressed as medians with interquartile ranges, unless otherwise indicated. Statistically significant differences between patients were determined using the Mann–Whitney U test for continuous data and Fisher’s exact test for categorical data. The association between the histologic findings and body composition indices was analyzed using the Spearman’s correlation coefficient. The association between advanced liver fibrosis and the body composition indices was analyzed using a logistic regression model. A receiver-operating characteristic (ROC) analysis for the body composition indices was performed to estimate the best cutoff values (COVs), calculated based on Youden’s index to predict advanced fibrosis. Any *p*-values < 0.05 were considered to be statistically significant. Statistical analyses were performed using R version 3.2.2 (R Foundation for Statistical Computing, Vienna, Austria).

## 3. Results

### 3.1. Baseline Characteristics

Of the 222 patients, 44 were excluded due to unavailability of CT scan images or histologic evidence of NAFLD (Appendix A). The baseline characteristics of the remaining 178 patients with biopsy-proven NAFLD are summarized in Table 1. The median age of the patients was 53.5 years (interquartile range (IQR), 38.0–64.0 years), and 86 patients (48.3%) were male. On the histologic profiles, the median NAS was 4.0 (IQR, 3.0–5.0). Among the 131 (73.6%) patients diagnosed with NASH, 61 (34.3%) had severe NASH. Furthermore, 47 (26.4%) and 32 (18.0%) patients had advanced liver fibrosis and cirrhosis, respectively. The median SMI, SATI, and VATI were 51.0 cm^2^/m^2^ (IQR, 45.8–58.2 cm^2^/m^2^), 64.4 cm^2^/m^2^ (IQR, 48.1–86.1 cm^2^/m^2^), and 66.7 cm^2^/m^2^ (IQR, 50.2–86.1 cm^2^/m^2^), respectively.

### 3.2. Skeletal Muscle and Adipose Tissue Distribution According to Severe Non-Alcoholic Steatohepatitis and Advanced Fibrosis

A comparison among the SMI, SATI, and VATI according to severe NASH or advanced fibrosis is shown in Figure 1. A comparative analysis of severe NASH revealed that the SMI (51.5 vs. 50.2 cm^2^/m^2^, *p* = 0.912) and SATI (62.8 vs. 65.9 cm^2^/m^2^, *p* = 0.216) were not significantly different between severe and non-severe NASH. However, the VATI was significantly greater in patients with severe NASH than in those without severe NASH (63.2 vs. 68.8 cm^2^/m^2^, *p* = 0.019). A comparative analysis of advanced fibrosis revealed that the SMI (51.6 vs. 49.9 cm^2^/m^2^, *p* = 0.140) and SATI (62.3 vs. 65.9 cm^2^/m^2^, *p* = 0.220) were not significantly different between advanced and non-advanced fibrosis. However, the VATI was significantly greater in patients with severe NASH than in those without severe NASH (62.8 vs. 74.1 cm^2^/m^2^, *p* < 0.001).

### 3.3. Correlation of Histologic Findings with Skeletal Muscle and Adipose Tissue Distribution

Next, we analyzed the correlation of the SMI, SATI, and VATI with the histologic grades of NAFLD, including steatosis, lobular inflammation, and ballooning degeneration (Figure 2 and Appendix A). The SMI was positively correlated with hepatic steatosis (*r_s_* = 0.20, *p* = 0.01) and negatively correlated with lobular inflammation (*r_s_* = −0.15, *p* = 0.04), but it was not with ballooning degeneration (*r_s_* = −0.06, *p* = 0.40). The SATI was positively correlated with hepatic steatosis (*r_s_* = 0.17, *p* = 0.02) and lobular inflammation (*r_s_* = 0.19, *p* = 0.01), but it was not with ballooning degeneration (*r_s_* = 0.08, *p* = 0.300). Although the VATI was not correlated with hepatic steatosis (*r_s_* = 0.13, *p* = 0.090), it was positively correlated with lobular inflammation (*r_s_* = 0.23, *p* < 0.001) and ballooning degeneration (*r_s_* = 0.26, *p* < 0.001). Additionally, we analyzed the correlation among the SMI, SATI, and VATI, the NAS, and the fibrosis stage (Figure 3). The SMI was not correlated with either the NAS (*r_s_* = 0.03, *p* = 0.680) or the fibrosis stage (*r_s_* = −0.13, *p* = 0.090). The SATI was positively correlated with the NAS (*r_s_* = 0.22, *p* < 0.001) but not with the fibrosis stage (*r_s_* = 0.13, *p* = 0.090). However, the VATI was positively correlated with both the NAS (*r_s_* = 0.30, *p* < 0.001) and the fibrosis stage (*r_s_* = 0.27, *p* < 0.001), which reflects the histologic severity of NAFLD.

### 3.4. Association of Skeletal Muscle and Adipose Tissue Distribution with Advanced Fibrosis

Before the multivariate analysis, we performed an ROC analysis of the VATI for advanced fibrosis to establish the optimal COVs (Appendix A). The areas under the ROC curves (AUCs) of the VATI for advanced fibrosis in male and female patients were 0.822 and 0.658, respectively. The optimal COVs of the VATI for male and female patients were 93.96 and 64.02 cm^2^/m^2^, respectively. All of the AUCs of the SMI and SATI were <0.6, suggesting poor discrimination in interpreting the AUC value, and validated COVs for the SATI were not observed. Therefore, we adjusted the mean COVs for the SATI, which was 65.5 cm^2^/m^2^ in male and 77.1 cm^2^/m^2^ in female patients. The COVs of the SMI were adjusted with the previously described values [24]. After converting the SMI, SATI, and VATI with categorical variables, age (OR, 1.09; 95% CI, 1.02–1.19; *p* = 0.025), platelet counts (OR, 0.98; 95% CI, 0.96–0.99; *p* < 0.001), severe NASH (OR, 8.66; 95% CI, 2.13–46.40; *p* = 0.005), and a high VATI (OR, 6.77; 95% CI, 1.81–29.90; *p* = 0.007) were found to be independent risk factors for advanced fibrosis in a multivariate analysis (Figure 4).

The presence of diabetes mellitus and hypertension tended to be independent risk factors for advanced fibrosis but were not statistically significant. (Table 2) In addition, these results were consistent in multivariate analysis with multiple adjusted models (Appendix A).

## 4. Discussion

In this study, cross-sectional images of CT scans were analyzed to determine the association between adipose tissue and skeletal muscle distribution and the histologic severity of NAFLD. Unlike the SMI and SATI, the VATI significantly increased in patients with severe NASH or advanced fibrosis. Moreover, the VATI is associated with both the NAS and the fibrosis stage, which is independently associated with advanced fibrosis.

The mechanisms regarding the association between sarcopenia and NAFLD are related to IR, chronic inflammation, oxidative stress, vitamin D deficiency, low physical activity, central obesity, and crosstalk with hepatokines (fibroblast growth factor 21, leukocyte cell-derived chemotaxin-2, and hepassocin) and myokines (irisin and myostatin) [27]. Although several studies have reported on the association between sarcopenia and NAFLD, most of these studies determined this association using noninvasive methods such as ultrasounds, the fatty liver index or the hepatic steatosis index for NAFLD, and bioimpedance analysis (BIA) for sarcopenia [28,29]. Specifically, all of the studies on patients with biopsy-proven NAFLD have evaluated sarcopenia using BIA, which leads to a possible discordance of the actual skeletal muscle mass [30]. Particularly, the discordance between CT and BIA can be frequently observed in adults aged > 65 years, female patients, and individuals with a BMI < 25 kg/m^2^ [30]. Therefore, according to this study, it would be appropriate to determine the skeletal muscle mass using CT when a high proportion of cases with advanced NAFLD are enrolled. Here, a low SMI was not associated with NAFLD severity and was negatively associated with lobular inflammation, a result inconsistent with the results of previous studies [17]. However, when analysis was performed in obese patients, the SMI was negatively associated with the fibrosis stage (*r_s_* = −0.18, *p* = 0.050, Appendix A). When sarcopenia and obesity are combined, the so-called sarcopenic obesity, sarcopenia-related factors may aggravate obesity-related diseases. An excessive fat accumulation induces the accumulation of ectopic fat in the skeletal muscle, which causes mitochondrial dysfunction and IR and creates a pro-inflammatory environment [31]. Thus, we believe that skeletal muscle loss was associated with lobular inflammation of the liver and commitment obesity acts synergistically, subsequently affecting NAFLD progression.

Activated adipocytes with hypertrophy and hyperplasia beyond their intrinsic limits induce the accumulation of pro-inflammatory macrophages and dysregulated immune cells, which produce various adipokines, immune cell-released cytokines, and chemokines [31]. Recently, the crosstalk between adipose tissue IR and liver macrophage has been considered to be a possible mechanism of NAFLD progression [32]. Unlike subcutaneous adiposity, visceral adiposity is more closely associated with metabolic diseases and adverse outcomes [9]. In this study, the SATI was significantly associated with steatosis grade, but the VATI was not. However, the SATI was not associated with ballooning degeneration and the fibrosis stage, but the VATI was. Adipose tissue distribution is dependent on sex in NAFLD. In a study of adolescent NAFLD patients, only visceral adiposity was associated with the severity of steatosis in male patients [33]. VAT adipocytes have a greater IR and a higher expression of androgen receptors, adiponectin, and IL-6 release than SAT [34]. However, unlike in a previous study on adolescents, the VATI was associated with the severity of steatosis in women with NAFLD only in our subgroup analysis (Appendix A). Here, the female patients were older than the male patients (44.5 ± 18.7 vs. 56.5 ± 11.8 years, Appendix A). We believe that age and sex affect fat accumulation in the liver. Thus, the discrepancy observed in the previous study is probably due to age and the postmenopausal status of woman.

In several studies, visceral adiposity was associated with the development [35,36] and severity [37,38,39] of NAFLD. In most of those studies, NAFLD diagnosis was established using a radiologic examination, such as ultrasound or CT, which has low sensitivity for detecting mild hepatic steatosis [35,36]. In some studies, the assessment of adipose tissue is established with the visceral adipose index by using several anthropometric and metabolic parameters, which have limited validation [35,37]. Although a few studies have been conducted on patients with biopsy-proven NAFLD, they had an inadequate sample size to reach conclusive outcomes [37,38,39]. Based on these findings, alteration of body composition is believed to be an important factor for NAFLD progression. A few studies have suggested that the skeletal muscle mass to visceral fat ratio, which is assessed to estimate sarcopenic obesity, is closely associated with MetS and cardiovascular diseases [40,41]. This simple index, used to assess sarcopenic obesity, was evaluated to determine NAFLD progression using transient elastography and controlled attenuation parameters in a few studies [41,42]. Although this index has not been analyzed in this study due to the small number of patients with sarcopenic obesity (*n* = 8, data not shown), it would be a potential risk factor when determining NAFLD progression based on our subgroup analysis (Appendix A).

This study has some limitations. First, there may be selection bias due to the enrollment of patients with possible severe disease who underwent a liver biopsy. Second, the laboratory tests regarding IR, such as homeostasis model assessment-IR and adipose tissue IR, were not analyzed due to the retrospective design of this study, with significant missing values. Third, the prognosis of NAFLD was not analyzed due to the cross-sectional design of this study. However, the liver fibrosis stage, which is closely associated with NAFLD mortality, was analyzed to estimate the prognosis of NAFLD [43]. Fourth, because all of the patients in this study are Asian, these results cannot be generalized in a Western population. Thus, further evaluation is required in a Western population. However, this study has the following strengths: both skeletal muscle mass and adipose tissue distribution were evaluated in biopsy-proven NAFLD with an adequate sample size and a comprehensive analysis of the histologic review.

In conclusion, the histologic severity of NAFLD is correlated with visceral adiposity. Visceral adiposity is independently associated with advanced fibrosis in patients with NAFLD. The measurement of visceral adiposity may be helpful to assess the risk of advanced liver disease in patients with NAFLD.

## Figures and Tables

**Figure 1 diagnostics-11-01061-f001:**
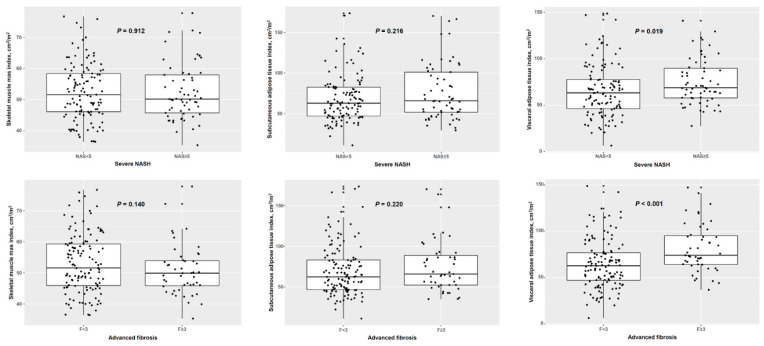
Adipose tissue distribution and skeletal muscle mass according to severe non-alcoholic steatohepatitis and advanced fibrosis. NASH, non-alcoholic steatohepatitis.

**Figure 2 diagnostics-11-01061-f002:**
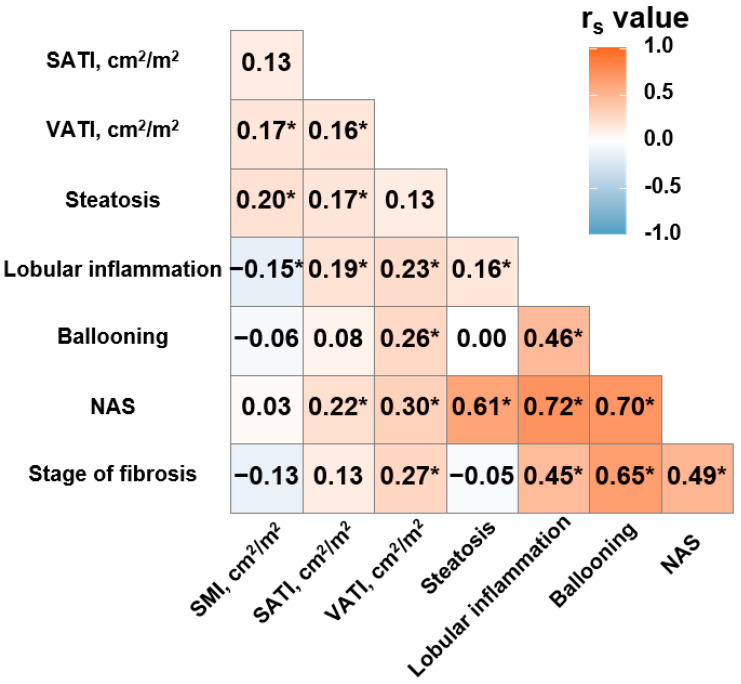
Correlation heatmap of pathologic findings, adipose tissue distribution and skeletal muscle mass. SMI, skeletal muscle index; SATI, subcutaneous adipose tissue index; VATI, visceral adipose tissue index; NAS, NAFLD activity score. Asterisk means *p* < 0.05. The heatmap data were analyzed using Spearman’s correlation.

**Figure 3 diagnostics-11-01061-f003:**
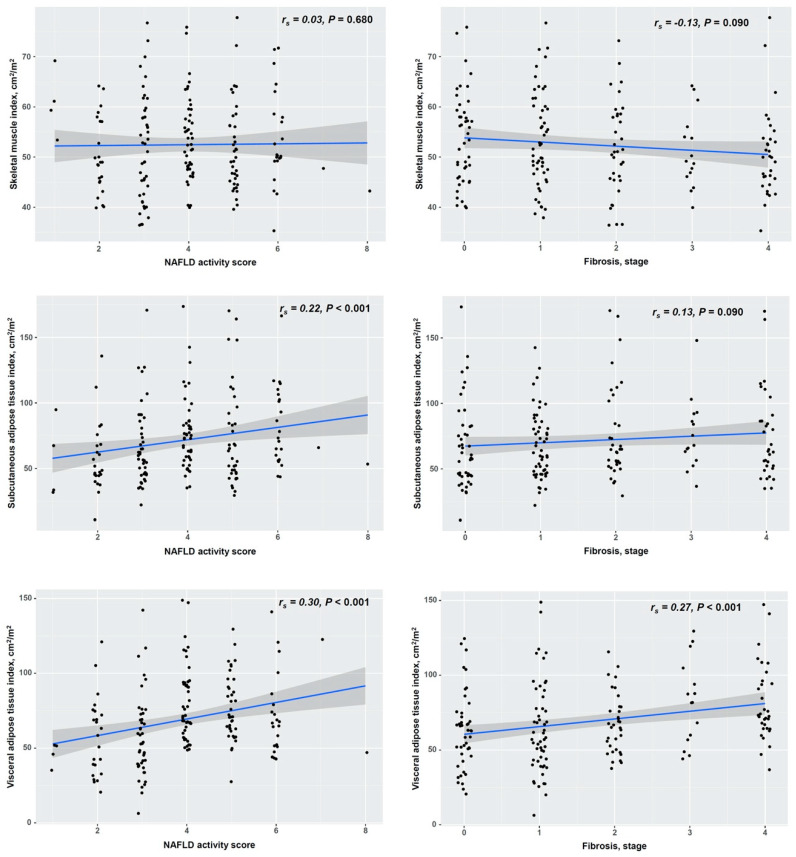
Correlation of adipose tissue distribution and skeletal muscle mass with the non-alcoholic fatty liver disease activity score and fibrosis stage. NAFLD, non-alcoholic fatty liver disease. The data were analyzed using Spearman’s correlation.

**Figure 4 diagnostics-11-01061-f004:**
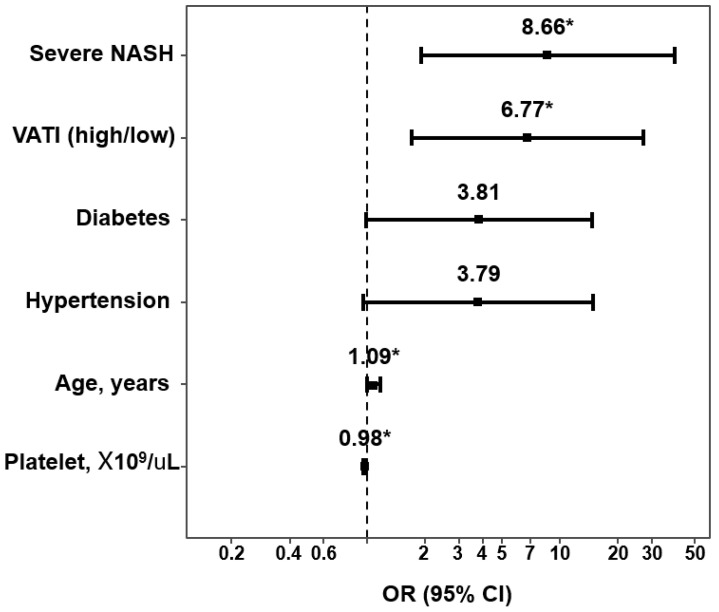
Risk factors for advanced fibrosis in patients with biopsy-proven non-alcoholic fatty liver disease in multivariate analysis. NASH, non-alcoholic fatty liver disease; VATI, visceral adipose tissue index. Asterisk means *p* < 0.05.

**Table 1 diagnostics-11-01061-t001:** Baseline characteristics.

Variable	Biopsy-Proven NAFLD Patients *n* = 178
Age, year	53.5 [38.0–64.0]
Male, *n* (%)	86 (48.3)
Body mass index, kg/m^2^	26.8 [24.4–29.5]
Comorbidities, *n* (%)	
Obesity	123 (69.1)
Diabetes mellitus	54 (30.7)
Hypertension	60 (34.1)
Liver function profiles	
Aspartate aminotransferase, IU/L	62.0 [43.0–96.0]
Alanine aminotransferase, IU/L	78.5 [43.0–96.0]
Platelet count, ×10^9^/uL	216.5 [179.0–274.0]
Gamma-glutamyl transferase, IU/L	62.5 [36.0–100.0]
Serum albumin, g/dL	4.4 [4.1–4.7]
Prothrombin time, INR	1.1 [1.0–1.1]
Metabolic profiles	
Fasting plasma glucose, mg/dL	108.5 [96.0–123.5]
Total cholesterol, mg/dL	177.5 [158.0–210.0]
Triglyceride, mg/dL	146.0 [106.5–223.5]
High-density lipoprotein, mg/dL	45.8 [35.0–53.2]
Low-density lipoprotein, mg/dL	99.7 [78.5–133.0]
C-reactive protein, mg/dL	0.2 [0.1–0.3]
Biopsy profiles	
NAFLD activity score	4.0 [3.0–5.0]
NAFLD activity score ≥ 5, *n* (%)	61 (34.3)
NASH, *n* (%)	131 (73.6)
Severe NASH, *n* (%)	61 (34.3)
Fibrosis stage, 0/1/2/3/4, *n* (%)	44/53/34/15/32 (24.7/29.8/19.1/8.4/18.0)
Advanced liver fibrosis, *n* (%)	47 (26.4)
Cirrhosis, *n* (%)	32 (18.0)
Muscle and fat distribution based on CT	
Skeletal muscle mass cm^2^/m^2^	51.0 [45.8–58.2]
Subcutaneous adipose tissue index cm^2^/m^2^	64.4 [48.1–86.1]
Visceral adipose tissue index cm^2^/m^2^	66.7 [50.2–86.1]

Values are expressed as median (interquartile range (IQR)) or *n* (%). NAFLD, non-alcoholic fatty liver disease; INR, international normalized ratio; NASH, non-alcoholic steatohepatitis; CT, computed tomography.

**Table 2 diagnostics-11-01061-t002:** Univariate and multivariate analysis using categorical body composition for advanced liver fibrosis in patients with non-alcoholic fatty liver disease.

Parameter	Univariate			Multivariate		
	OR	95% CI	*p*-Value	OR	95% CI	*p*-Value
Age, years	1.12	1.08–1.16	<0.001	1.09	1.02–1.19	0.025
Male (yes/no)	3.79	1.80–7.94	<0.001			
Obesity (yes/no)	1.23	0.59–2.58	0.576			
Diabetes mellitus (yes/no)	4.40	2.14–9.02	<0.001	3.81	1.04–15.98	0.051
Hypertension (yes/no)	5.85	2.81–12.15	<0.001	3.79	0.99–15.89	0.056
ALT, U/L	0.98	0.98–0.99	0.002			
GGT, U/L	1.00	1.00–1.00	0.753			
PLT, ×10^9^/L	0.97	0.97–0.98	<0.001	0.98	0.96–0.99	<0.001
Albumin, g/dL	1.06	0.94–1.19	0.360			
PT INR	0.91	0.59–1.41	0.669			
CRP, mg/dL	0.76	0.37–1.56	0.458			
HDL, mg/dL	0.98	0.95–1.01	0.187			
LDL, mg/dL	0.98	0.97–1.00	0.005			
TG, mg/dL	0.99	0.99–1.00	0.020			
Severe NASH (yes/no)	3.85	1.92–7.74	<0.001	8.66	2.13–46.40	0.005
SMI (high/low)	0.23	0.05–1.03	0.055			
SATI (high/low)	1.13	0.57–2.21	0.732			
VATI (high/low)	6.90	3.28–14.54	<0.001	6.77	1.81–29.90	0.007

OR, odds ratio; CI, confidence interval; ALT, alanine aminotransferase; GGT, gamma-glutamyl transferase; PLT, platelet counts; PT INR, prothrombin time international normalized ratio; CRP, c-reactive protein; HDL, high-density lipoprotein; LDL, low-density lipoprotein; TG, triglyceride; NASH, non-alcoholic steatohepatitis; SMI, skeletal muscle index; SATI, subcutaneous adipose tissue index; VATI, visceral adipose tissue index.

## Data Availability

The data that support the findings of this study are also available from the corresponding authors (J.G.P. and S.Y.P.) upon reasonable request.

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
