# Peer review of "Association of Skeletal Muscle and Adipose Tissue Distribution with Histologic Severity of Non-Alcoholic Fatty Liver"

_diagnostics, 2021, doi:10.3390/diagnostics11061061_

Round 1

Reviewer 1 Report

General comments:

Statistical analysis paragraph is very concise. In order to improve the readability of the manuscript, please consider to add, at least in the caption of figures, a better description of variable and statistical method used.

Please consider to have also caption and figure in the same page.

There are some problems in figure 2: why are there different variables in horizontal and vertical axes?  why SMI and SATI variables do not correlate with themselves? 

Figure 3: do you add the boxplot into the variable distribution? in this way figures will be more informative. Same in supplementary figure S3, S5, S6

Figure 4: the figure is confusing. please describe horizontal axis and scale used, the meaning of the asterisk and the meaning of the dashed line.
Please consider to add text between 212-218 lines into the figure4 caption.

Author Response

Response to Reviewer 1 Comments

  1. Statistical analysis paragraph is very concise. In order to improve the readability of the manuscript, please consider to add, at least in the caption of figures, a better description of variable and statistical method used. Please consider to have also caption and figure in the same page.

Response 1.)

We really appreciate the careful reading of our manuscript. We appreciate the time and effort that you have dedicated to providing valuable comments on our manuscript. I fully agree with your opinions, and additional comments has been inserted. As you commented, the caption of each figure was modified, and statistical techniques were described in some figures to improve readability. Figure captions are changed as follows.

Figure 1. Adipose tissue distribution and skeletal muscle mass according to severe non-alcoholic steatohepatitis and advanced fibrosis. NASH, non-alcoholic steatohepatitis

Figure 2. Correlation heatmap of pathologic findings, adipose tissue distribution and skeletal muscle mass. SMI, skeletal muscle index; SATI, subcutaneous adipose tissue in-dex; VATI, visceral adi-pose tissue index; NAS, NAFLD activity score. Asterisk means p < 0.05. The heatmap data were analyzed using Spearman's correlation.

Figure 3. Correlation of adipose tissue distribution and skeletal muscle mass with the non-alcoholic fatty liver disease activity score and fibrosis stage. NAFLD, non-alcoholic fatty liver disease. The data were analyzed using Spearman's correlation.

Figure 4. Risk factors for advanced fibrosis in patients with biopsy proven non-alcoholic fatty liver disease in multivariate analysis. NASH, non-alcoholic fatty liver disease; VATI, viceral adipose tissue index. Asterisk means p < 0.05.

  1. There are some problems in figure 2: why are there different variables in horizontal and vertical axes? why SMI and SATI variables do not correlate with themselves?

Response 2)

Thank you so much for finding the fatal flaw. We apologize for the confusion due to the misspelling of variable. The heatmap was modified so that the Rs and p value according to each variable could be identified.

We've listed so easy to see that p <0.05 using an asterisk. Thanks again for your meticulous review.

  1. Figure 3: do you add the boxplot into the variable distribution? in this way figures will be more informative. Same in supplementary figure S3, S5, S6

Response 3)

Thanks for the good comments. We considered each BOX PLOT GRAPH, but we ask for your understanding that it is not included because the number of people in each group is too small and the distribution is diverse. Instead, the rs value was shown to indicate the overall trend according to the change of the overall grade.

  1. Figure 4: the figure is confusing. please describe horizontal axis and scale used, the meaning of the asterisk and the meaning of the dashed line.

Reponse 4)

Thanks for the good comments. We've listed so easy to recognize that p <0.05 using an asterisk. It has been modified by adding OR (95% CI) as follows.

5. Please consider to add text between 212-218 lines into the figure4 caption.

Response 5)

I agree with your opinion, and inserted the following text between the table and the figure.

The presence of diabetes mellitus and hypertension tended tobe independent risk facotrs for advanced fibrosis, but were not statistically significant. (Table 2) In addition, these results were consistent in multivariate analysis with multiple models adjusted (Table S1).

Thanks again for your kind and valuable comments. We hope that our revision will meet with approval. We would like to respond to any further questions and comments you may have.

Reviewer 2 Report

The Authors conducted a cross-sectional study on Korean patients with biopsy-proven NAFLD aiming to investigate the association of the skeletal muscle mass and adipose tissue distribution using CT with histologic severity of NAFLD.

The study was properly designed. Methods were adequately described.

Visceral adiposity was found independently associated with advanced fibrosis in NAFLD patients. Results were clearly presented. 

The Discussion section need to be separated from verse 219. 

Author Response

Response to Reviewer 2 Comments

The Authors conducted a cross-sectional study on Korean patients with biopsy-proven NAFLD aiming to investigate the association of the skeletal muscle mass and adipose tissue distribution using CT with histologic severity of NAFLD.

The study was properly designed. Methods were adequately described.

Visceral adiposity was found independently associated with advanced fibrosis in NAFLD patients. Results were clearly presented.

The Discussion section need to be separated from verse 219

Response)

Thank you for careful reading of our manuscript. We appreciate the time and effort that you have dedicated to providing valuable comments on our manuscript. We discovered that the discussion title was removed from your meticulous reviews. (line 240)

Thanks again for your kind and valuable comments. We hope that our revision will meet with approval. We would like to respond to any further questions and comments you may have.